# Semi-Supervised Learning for Optical Flow
# with Generative Adversarial Networks

**Wei-Sheng Lai**[1]          **Jia-Bin Huang**[2]          **Ming-Hsuan Yang**[1,3]
[1]University of California, Merced     [2]Virginia Tech     [3]Nvidia Research
[1]{wlai24|mhyang}@ucmerced.edu               [2]jbhuang@vt.edu

## Abstract

Convolutional neural networks (CNNs) have recently been applied to the optical flow estimation problem. As training the CNNs requires sufficiently large amounts of labeled data, existing approaches resort to synthetic, unrealistic datasets. On the other hand, unsupervised methods are capable of leveraging real-world videos for training where the ground truth flow fields are not available. These methods, however, rely on the fundamental assumptions of brightness constancy and spatial smoothness priors that do not hold near motion boundaries. In this paper, we propose to exploit unlabeled videos for semi-supervised learning of optical flow with a Generative Adversarial Network. Our key insight is that the adversarial loss can capture the structural patterns of flow warp errors without making explicit assumptions. Extensive experiments on benchmark datasets demonstrate that the proposed semi-supervised algorithm performs favorably against purely supervised and baseline semi-supervised learning schemes.

## 1   Introduction

Optical flow estimation is one of the fundamental problems in computer vision. The classical formulation builds upon the assumptions of brightness constancy and spatial smoothness [15, 25]. Recent advancements in this field include using sparse descriptor matching as guidance [4], leveraging dense correspondences from hierarchical features [2, 39], or adopting edge-preserving interpolation techniques [32]. Existing classical approaches, however, involve optimizing computationally expensive non-convex objective functions.

With the rapid growth of deep convolutional neural networks (CNNs), several approaches have been proposed to solve optical flow estimation in an end-to-end manner. Due to the lack of the large-scale ground truth flow datasets of real-world scenes, existing approaches [8, 16, 30] rely on training on synthetic datasets. These synthetic datasets, however, do not reflect the complexity of realistic photometric effects, motion blur, illumination, occlusion, and natural image noise. Several recent methods [1, 40] propose to leverage real-world videos for training CNNs in an unsupervised setting (i.e., without using ground truth flow). The main idea is to use loss functions measuring brightness constancy and spatial smoothness of flow fields as a proxy for losses using ground truth flow. However, the assumptions of brightness constancy and spatial smoothness often do not hold near motion boundaries. Despite the acceleration in computational speed, the performance of these approaches still does not match up to the classical flow estimation algorithms.

With the limited quantity and unrealistic of ground truth flow and the large amounts of real-world unlabeled data, it is thus of great interest to explore the semi-supervised learning framework. A straightforward approach is to minimize the End Point Error (EPE) loss for data with ground truth flow and the loss functions that measure classical brightness constancy and smoothness assumptions for unlabeled training images (Figure 1 (a)). However, we show that such an approach is sensitive to the choice of parameters and may sometimes decrease the accuracy of flow estimation. Prior work [1, 40] minimizes a robust loss function (e.g., Charbonnier function) on the flow warp error

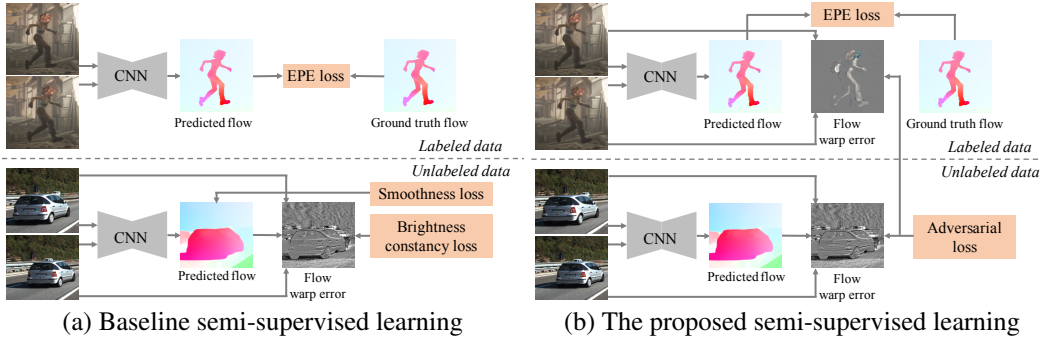

(a) Baseline semi-supervised learning          (b) The proposed semi-supervised learning

Figure 1: **Semi-supervised learning for optical flow estimation.** (a) A baseline semi-supervised algorithm utilizes the assumptions of brightness constancy and spatial smoothness to train CNN from unlabeled data (e.g., [1, 40]). (b) We train a generative adversarial network to capture the structure patterns in flow warp error images without making any prior assumptions.

(i.e., the difference between the first input image and the warped second image) by modeling the brightness constancy with a Laplacian distribution. As shown in Figure 2, although robust loss functions can fit the likelihood of the *per-pixel* flow warp error well, the spatial structure in the warp error images cannot be modeled by simple distributions. Such structural patterns often arise from occlusion and dis-occlusion caused by large object motion, where the brightness constancy assumption does not hold. A few approaches have been developed to cope with such *brightness inconsistency* problem using the Fields-of-Experts (FoE) [37] or a Gaussian Mixture Model (GMM) [33]. However, the inference of optical flow entails solving time-consuming optimization problems.

In this work, our goal is to leverage both the labeled and the unlabeled data *without* making explicit assumptions on the brightness constancy and flow smoothness. Specifically, we propose to impose an adversarial loss [12] on the flow warp error image to replace the commonly used brightness constancy loss. We formulate the optical flow estimation as a conditional Generative Adversarial Network (GAN) [12]. Our generator takes the input image pair and predicts the flow. We then compute the flow warp error image using a bilinear sampling layer. We learn a discriminator to distinguish between the flow warp error from predicted flow and ground truth optical flow fields. The adversarial training scheme encourages the generator to produce the flow warp error images that are indistinguishable from the ground truth. The adversarial loss serves as a regularizer for *both* labeled and unlabeled data (Figure 1 (b)). With the adversarial training, our network learns to model the structural patterns of flow warp error to refine the motion boundary. During the test phase, the generator can efficiently predict optical flow in one feed-forward pass.

We make the following three contributions:

- We propose a generative adversarial training framework to learn to predict optical flow by leveraging both labeled and unlabeled data in a semi-supervised learning framework.
- We develop a network to capture the spatial structure of the flow warp error without making primitive assumptions on brightness constancy or spatial smoothness.
- We demonstrate that the proposed semi-supervised flow estimation method outperforms the purely supervised and baseline semi-supervised learning when using the same amount of ground truth flow and network parameters.

## 2   Related Work

In the following, we discuss the learning-based optical flow algorithms, CNN-based semi-supervised learning approaches, and generative adversarial networks within the context of this work.

**Optical flow.**   Classical optical flow estimation approaches typically rely on the assumptions of brightness constancy and spatial smoothness [15, 25]. Sun et al. [36] provide a unified review of classical algorithms. Here we focus our discussion on recent learning-based methods in this field.

Learning-based methods aim to learn priors from natural image sequences without using hand-crafted assumptions. Sun et al. [37] assume that the flow warp error at each pixel is *independent* and use a set of linear filters to learn the brightness inconsistency. Rosenbaum and Weiss [33] use a GMM to learn the flow warp error at the *patch* level. The work of Rosenbaum et al. [34] learns patch priors

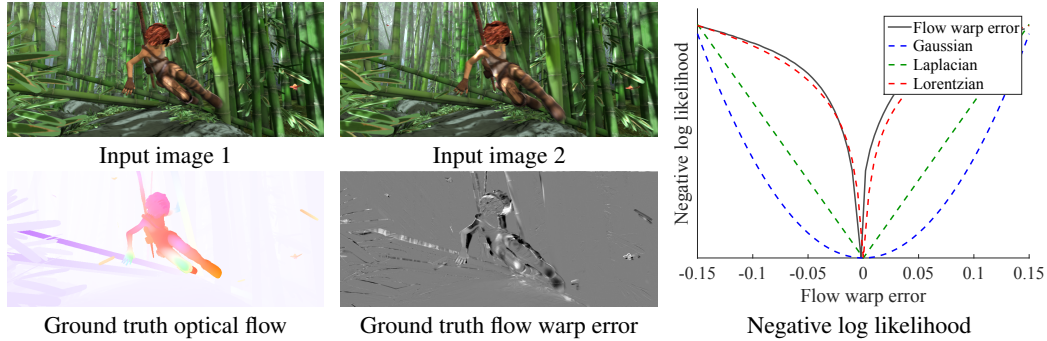

| Input image 1 | Input image 2 | |
| Ground truth optical flow | Ground truth flow warp error | Negative log likelihood |

Figure 2: **Modeling the distribution of flow warp error.** The robust loss functions, e.g., Lorentzian or Charbonnier functions, can model the distribution of per-pixel flow warp error well. However, the spatial pattern resulting from large motion and occlusion cannot be captured by simple distributions.

to model the local flow statistics. These approaches incorporate the learned priors into the classical formulation and thus require solving time-consuming alternative optimization to infer the optical flow. Furthermore, the limited amount of training data (e.g., Middlebury [3] or Sintel [5]) may not fully demonstrate the capability of learning-based optical flow algorithms. In contrast, we train a deep CNN with large datasets (FlyingChairs [8] and KITTI [10]) in an end-to-end manner. Our model can predict flow efficiently in a single feed-forward pass.

The FlowNet [8] presents a deep CNN approach for learning optical flow. Even though the network is trained on a large dataset with ground truth flow, strong data augmentation and the variational refinement are required. Ilg et al. [16] extend the FlowNet by stacking multiple networks and using more training data with different motion types including complex 3D motion and small displacements. To handle large motion, the SPyNet approach [30] estimates flow in a classical spatial pyramid framework by warping one of the input images and predicting the residual flow at each pyramid level.

A few attempts have recently been made to learn optical flow from unlabeled videos in an unsupervised manner. The USCNN method [1] approximates the brightness constancy with a Taylor series expansion and trains a deep network using the UCF101 dataset [35]. Yu et al. [40] enables the back-propagation of the warping function using the bilinear sampling layer from the spatial transformer network [18] and explicitly optimizes the brightness constancy and spatial smoothness assumptions. While Yu et al. [40] demonstrate comparable performance with the FlowNet on the KITTI dataset, the method requires significantly more sophisticated data augmentation techniques and different parameter settings for each dataset. Our approach differs from these methods in that we use both labeled and unlabeled data to learn optical flow in a semi-supervised framework.

**Semi-supervised learning.** Several methods combine the classification objective with unsupervised reconstruction losses for image recognition [31, 41]. In low-level vision tasks, Kuznietsov et al. [21] train a deep CNN using sparse ground truth data for single-image depth estimation. This method optimizes a supervised loss for pixels with ground truth depth value as well as an unsupervised image alignment cost and a regularization cost. The image alignment cost resembles the brightness constancy, and the regularization cost enforces the spatial smoothness on the predicted depth maps. We show that adopting a similar idea to combine the EPE loss with image reconstruction and smoothness losses may not improve flow accuracy. Instead, we use the adversarial training scheme for learning to model the structural flow warp error without making assumptions on images or flow.

**Generative adversarial networks.** The GAN framework [12] has been successfully applied to numerous problems, including image generation [7, 38], image inpainting [28], face completion [23], image super-resolution [22], semantic segmentation [24], and image-to-image translation [17, 42].

Within the scope of domain adaptation [9, 14], the discriminator learns to differentiate the features from the two different domains, e.g., synthetic, and real images. Koziński et al. [20] adopt the adversarial training framework for semi-supervised learning on the image segmentation task where the discriminator is trained to distinguish between the predictions produced from labeled and unlabeled data. Different from Koziński et al. [20], our discriminator learns to distinguish the flow warp errors between using the ground truth flow and using the estimated flow. The generator thus learns to model the spatial structure of flow warp error images and can improve flow estimation accuracy around motion boundaries.

## 3 Semi-Supervised Optical Flow Estimation

In this section, we describe the semi-supervised learning approach for optical flow estimation, the design methodology of the proposed generative adversarial network for learning the flow warp error, and the use of the adversarial loss to leverage labeled and unlabeled data.

### 3.1 Semi-supervised learning

We address the problem of learning optical flow by using both labeled data (i.e., with the ground truth dense optical flow) and unlabeled data (i.e., raw videos). Given a pair of input images $\{I_1, I_2\}$, we train a deep network to generate the dense optical flow field $f = [u, v]$. For labeled data with the ground truth optical flow (denoted by $\hat{f} = [\hat{u}, \hat{v}]$), we optimize the EPE loss between the predicted and ground truth flow:

$$\mathcal{L}_{\text{EPE}}(f, \hat{f}) = \sqrt{(u - \hat{u})^2 + (v - \hat{v})^2}. \tag{1}$$

For unlabeled data, existing work [40] makes use of the classical brightness constancy and spatial smoothness to define the image warping loss and flow smoothness loss:

$$\mathcal{L}_{\text{warp}}(I_1, I_2, f) = \rho\left(I_1 - \mathbb{W}\left(I_2, f\right)\right), \tag{2}$$
$$\mathcal{L}_{\text{smooth}}(f) = \rho(\partial_x u) + \rho(\partial_y u) + \rho(\partial_x v) + \rho(\partial_y v), \tag{3}$$

where $\partial_x$ and $\partial_y$ are horizontal and vertical gradient operators and $\rho(\cdot)$ is the robust penalty function. The warping function $\mathbb{W}(I_2, f)$ uses the bilinear sampling [18] to warp $I_2$ according to the flow field $f$. The difference $I_1 - \mathbb{W}(I_2, f)$ is the flow warp error as shown in Figure 2. Minimizing $\mathcal{L}_{\text{warp}}(I_1, I_2, f)$ enforces the flow warp error to be close to zero at *every* pixel.

A baseline semi-supervised learning approach is to minimize $\mathcal{L}_{\text{EPE}}$ for labeled data and minimize $\mathcal{L}_{\text{warp}}$ and $\mathcal{L}_{\text{smooth}}$ for unlabeled data:

$$\sum_{i \in D_l} \mathcal{L}_{\text{EPE}}\left(f^{(i)}, \hat{f}^{(i)}\right) + \sum_{j \in D_u} \left(\lambda_w \mathcal{L}_{\text{warp}}\left(I_1^{(j)}, I_2^{(j)}, f^{(j)}\right) + \lambda_s \mathcal{L}_{\text{smooth}}\left(f^{(j)}\right)\right), \tag{4}$$

where $D_l$ and $D_u$ represent labeled and unlabeled datasets, respectively. However, the commonly used robust loss functions (e.g., Lorentzian and Charbonnier) assume that the error is independent at each pixel and thus cannot model the structural patterns of flow warp error caused by occlusion. Minimizing the combination of the supervised loss in (1) and unsupervised losses in (2) and (3) may degrade the flow accuracy, especially when large motion present in the input image pair. As a result, instead of using the unsupervised losses based on classical assumptions, we propose to impose an adversarial loss on the flow warp images within a generative adversarial network. We use the adversarial loss to regularize the flow estimation for both labeled and unlabeled data.

### 3.2 Adversarial training

Training a GAN involves optimizing the two networks: a generator $G$ and a discriminator $D$. The generator $G$ takes a pair of input images to generate optical flow. The discriminator $D$ performs binary classification to distinguish whether a flow warp error image is produced by the estimated flow from the generator $G$ or by the ground truth flow. We denote the flow warp error image from the ground truth flow and generated flow by $\hat{y} = I_1 - \mathbb{W}(I_2, \hat{f})$ and $y = I_1 - \mathbb{W}(I_2, f)$, respectively. The objective function to train the GAN can be expressed as:

$$\mathcal{L}_{\text{adv}}(y, \hat{y}) = \mathbb{E}_{\hat{y}}[\log D(\hat{y})] + \mathbb{E}_y[\log\left(1 - D(y)\right)]. \tag{5}$$

We incorporate the adversarial loss with the supervised EPE loss and solve the following minmax problem for optimizing $G$ and $D$:

$$\min_G \max_D \mathcal{L}_{\text{EPE}}(G) + \lambda_{\text{adv}} \mathcal{L}_{\text{adv}}(G, D), \tag{6}$$

where $\lambda_{\text{adv}}$ controls the relative importance of the adversarial loss for optical flow estimation.

Following the standard procedure for GAN training, we alternate between the following two steps to solve (6): (1) update the discriminator $D$ while holding the generator $G$ fixed and (2) update generator $G$ while holding the discriminator $D$ fixed.

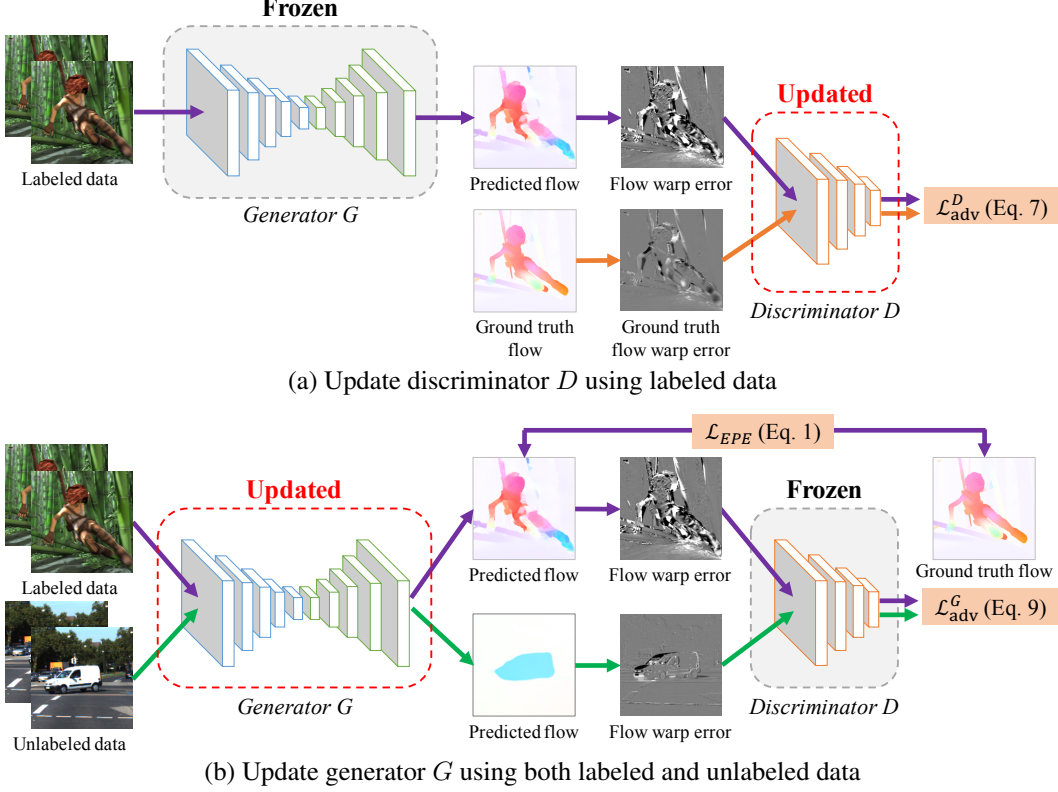

(a) Update discriminator $D$ using labeled data

(b) Update generator $G$ using both labeled and unlabeled data

Figure 3: **Adversarial training procedure.** Training a generative adversarial network involves the alternative optimization of the discriminator $D$ and generator $G$.

**Updating discriminator $D$.** We train the discriminator $D$ to classify between the ground truth flow warp error (real samples, labeled as 1) and the flow warp error from the predicted flow (fake samples, labeled as 0). The maximization of (5) is equivalent to minimizing the binary cross-entropy loss $\mathcal{L}_{\text{BCE}}(p, t) = -t \log(p) - (1 - t) \log(1 - p)$ where $p$ is the output from the discriminator and $t$ is the target label. The adversarial loss for updating $D$ is defined as:

$$\mathcal{L}^D_{\text{adv}}(y, \hat{y}) = \mathcal{L}_{\text{BCE}}(D(\hat{y}), 1) + \mathcal{L}_{\text{BCE}}(D(y), 0)$$
$$= -\log D(\hat{y}) - \log(1 - D(y)). \tag{7}$$

As the ground truth flow is required to train the discriminator, only the labeled data $D_l$ is involved in this step. By fixing $G$ in (6), we minimize the following loss function for updating $D$:

$$\sum_{i \in D_l} \mathcal{L}^D_{\text{adv}}(y^{(i)}, \hat{y}^{(i)}). \tag{8}$$

**Updating generator $G$.** The goal of the generator is to "fool" the discriminator by producing flow to generate realistic flow warp error images. Optimizing (6) with respect to $G$ becomes minimizing $\log(1 - D(y))$. As suggested by Goodfellow et al. [12], one can instead minimize $-\log(D(y))$ to speed up the convergence. The adversarial loss for updating $G$ is then equivalent to the binary cross entropy loss that assigns label 1 to the generated flow warp error $y$:

$$\mathcal{L}^G_{\text{adv}}(y) = \mathcal{L}_{\text{BCE}}(D(y), 1) = -\log(D(y)). \tag{9}$$

By combining the adversarial loss with the supervised EPE loss, we minimize the following function for updating $G$:

$$\sum_{i \in D_l} \left( \mathcal{L}_{\text{EPE}}\left(f^{(i)}, \hat{f}^{(i)}\right) + \lambda_{\text{adv}} \mathcal{L}^G_{\text{adv}}(y^{(i)}) \right) + \sum_{j \in D_u} \lambda_{\text{adv}} \mathcal{L}^G_{\text{adv}}(y^{(j)}). \tag{10}$$

We note that the adversarial loss is computed for *both* labeled and unlabeled data, and thus guides the flow estimation for image pairs without the ground truth flow. Figure 3 illustrates the two main steps to update the generator $D$ and the discriminator $G$ in the proposed semi-supervised learning framework.

### 3.3 Network architecture and implementation details

**Generator.** We construct a 5-level SPyNet [30] as our generator. Instead of using simple stacks of convolutional layers as sub-networks [30], we choose the encoder-decoder architecture with skip connections to effectively increase the receptive fields. Each convolutional layer has a $3 \times 3$ spatial support and is followed by a ReLU activation. We present the details of our SPyNet architecture in the supplementary material.

**Discriminator.** As we aim to learn the local structure of flow warp error at motion boundaries, it is more effective to penalize the structure at the scale of local patches instead of the whole image. Therefore, we use the PatchGAN [17] architecture as our discriminator. The PatchGAN is a fully convolutional classifier that classifies whether each $N \times N$ overlapping patch is real or fake. The PatchGAN has a receptive field of $47 \times 47$ pixels.

**Implementation details.** We implement the proposed method using the Torch framework [6]. We use the Adam solver [19] to optimize both the generator and discriminator with $\beta_1 = 0.9$, $\beta_2 = 0.999$ and the weight decay of $1e - 4$. We set the initial learning rate as $1e - 4$ and then multiply by 0.5 every 100k iterations after the first 200k iterations. We train the network for a total of 600k iterations.

We use the FlyingChairs dataset [8] as the labeled dataset and the KITTI raw videos [10] as the unlabeled dataset. In each mini-batch, we randomly sample 4 image pairs from each dataset. We randomly augment the training data in the following ways: (1) *Scaling* between $[1, 2]$, (2) *Rotating* within $[-17°, 17°]$, (3) Adding *Gaussian noise* with a sigma of 0.1, (4) Using *color jitter* with respect to brightness, contrast and saturation uniformly sampled from $[0, 0.04]$. We then crop images to $384 \times 384$ patches and normalize by the mean and standard deviation computed from the ImageNet dataset [13]. The source code is publicly available on `http://vllab.ucmerced.edu/wlai24/semiFlowGAN`.

## 4 Experimental Results

We evaluate the performance of optical flow estimation on five benchmark datasets. We conduct ablation studies to analyze the contributions of individual components and present comparisons with the state-of-the-art algorithms including classical variational algorithms and CNN-based approaches.

### 4.1 Evaluated datasets and metrics

We evaluate the proposed optical flow estimation method on the benchmark datasets: MPI-Sintel [5], KITTI 2012 [11], KITTI 2015 [27], Middlebury [3] and the test set of FlyingChairs [8]. The MPI-Sintel and FlyingChairs are synthetic datasets with dense ground truth flow. The Sintel dataset provides two rendered sets, Clean and Final, that contain both small displacements and large motion. The training and test sets contain 1041 and 552 image pairs, respectively. The FlyingChairs test set is composed of 640 image pairs with similar motion statistics to the training set. The Middlebury dataset has only eight image pairs with small motion. The images from the KITTI 2012 and KITTI 2015 datasets are collected from driving real-world scenes with large forward motion. The ground truth optical flow is obtained from a 3D laser scanner and thus only covers about $50\%$ of image pixels. There are 194 image pairs in the KITTI 2012 dataset, and 200 image pairs in the KITTI 2015 dataset.

We compute the average EPE (1) on pixels with the ground truth flow available for each dataset. On the KITTI-2015 dataset, we also compute the Fl score [27], which is the ratio of pixels that have EPE greater than 3 pixels *and* $5\%$ of the ground truth value.

### 4.2 Ablation study

We conduct ablation studies to analyze the contributions of the adversarial loss and the proposed semi-supervised learning with different training schemes.

**Adversarial loss.** We adjust the weight of the adversarial loss $\lambda_{\text{adv}}$ in (10) to validate the effect of the adversarial training. When $\lambda_{\text{adv}} = 0$, our method falls back to the fully supervised learning setting. We show the quantitative evaluation in Table 1. Using larger values of $\lambda_{\text{adv}}$ may decrease the performance and cause visual artifacts as shown in Figure 4. We therefore choose $\lambda_{\text{adv}} = 0.01$.

Table 1: **Analysis on adversarial loss.** We train the proposed model using different weights for the adversarial loss in (10).

| $\lambda_{adv}$ | Sintel-Clean EPE | Sintel-Final EPE | KITTI 2012 EPE | KITTI 2015 EPE | KITTI 2015 Fl-all | FlyingChairs EPE |
|---|---|---|---|---|---|---|
| 0 | 3.51 | 4.70 | 7.69 | 17.19 | 40.82% | 2.15 |
| 0.01 | **3.30** | **4.68** | **7.16** | **16.02** | **38.77%** | **1.95** |
| 0.1 | 3.57 | 4.73 | 8.25 | 16.82 | 42.78% | 2.11 |
| 1 | 3.93 | 5.18 | 13.89 | 21.07 | 63.43% | 2.21 |

Table 2: **Analysis on receptive field of discriminator.** We vary the number of strided convolutional layers in the discriminator to achieve different size of receptive fields.

| # Strided convolutions | Receptive field | Sintel-Clean EPE | Sintel-Final EPE | KITTI 2012 EPE | KITTI 2015 EPE | KITTI 2015 Fl-all | FlyingChairs EPE |
|---|---|---|---|---|---|---|---|
| $d = 2$ | $23 \times 23$ | 3.66 | 4.90 | 7.38 | 16.28 | 40.19% | 2.15 |
| $d = 3$ | $47 \times 47$ | **3.30** | **4.68** | **7.16** | **16.02** | **38.77%** | **1.95** |
| $d = 4$ | $95 \times 95$ | 3.70 | 5.00 | 7.54 | 16.38 | 41.52% | 2.16 |

**Receptive fields of discriminator.** The receptive field of the discriminator is equivalent to the size of patches used for classification. The size of the receptive field is determined by the number of strided convolutional layers, denoted by $d$. We test three different values, $d = 2, 3, 4$, which are corresponding to the receptive field of $23 \times 23$, $47 \times 47$, and $95 \times 95$, respectively. As shown in Table 2, the network with $d = 3$ performs favorably against other choices on all benchmark datasets. Using too large or too small patch sizes might not be able to capture the structure of flow warp error well. Therefore, we design our discriminator to have a receptive field of $47 \times 47$ pixels.

**Training schemes.** We train the same network (i.e., our generator $G$) with the following training schemes: (a) *Supervised*: minimizing the EPE loss (1) on the FlyingChairs dataset. (b) *Unsupervised*: minimizing the classical brightness constancy (2) and spatial smoothness (3) using the Charbonnier loss function on the KITTI raw dataset. (c) *Baseline semi-supervised*: minimizing the combination of supervised and unsupervised losses (4) on the FlyingChairs and KITTI raw datasets. For the semi-supervised setting, we evaluate different combinations of $\lambda_w$ and $\lambda_s$ in Table 3. We note that it is not easy to run grid search to find the best parameter combination for all evaluated datasets. We choose $\lambda_w = 1$ and $\lambda_s = 0.01$ for the baseline semi-supervised and unsupervised settings.

We provide the quantitative evaluation of the above training schemes in Table 4 and visual comparisons in Figure 5 and 6. As images in KITTI 2015 have large forward motion, there are large occluded/dis-occluded regions, particularly on the image and moving object boundaries. The brightness constancy does not hold in these regions. Consequently, minimizing the image warping loss (2) results in inaccurate flow estimation. Compared to the fully supervised learning, our method further refines the motion boundaries by modeling the flow warp error. By incorporating both labeled and unlabeled data in training, our method effectively reduces EPEs on the KITTI 2012 and 2015 datasets.

**Training on partially labeled data.** We further analyze the effect of the proposed semi-supervised method by reducing the amount of labeled training data. Specifically, we use 75%, 50% and 25%

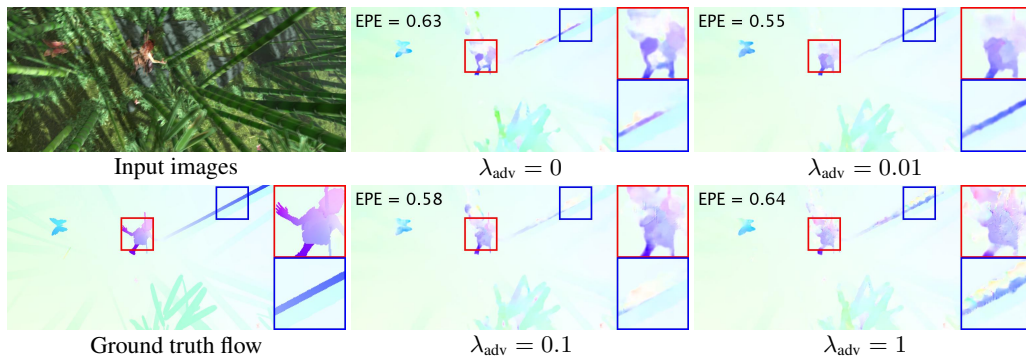

Figure 4: **Comparisons of adversarial loss $\lambda_{adv}$.** Using larger value of $\lambda_{adv}$ does not necessarily improve the performance and may cause unwanted visual artifacts.

Table 3: **Evaluation for baseline semi-supervised setting.** We test different combinations of $\lambda_w$ and $\lambda_s$ in (4). We note that it is difficult to find the best parameters for all evaluated datasets.

| $\lambda_w$ | $\lambda_s$ | Sintel-Clean EPE | Sintel-Final EPE | KITTI 2012 EPE | KITTI 2015 EPE | KITTI 2015 Fl-all | FlyingChairs EPE |
|---|---|---|---|---|---|---|---|
| 1 | 0 | 3.77 | 5.02 | 10.90 | 18.52 | 39.94% | 2.25 |
| 1 | 0.1 | 3.75 | 5.05 | 11.82 | 19.98 | 43.18% | 2.19 |
| 1 | 0.01 | 3.69 | 4.86 | 10.38 | **18.07** | **39.33**% | **2.11** |
| 0.1 | 0.01 | 3.64 | **4.81** | 10.15 | 18.94 | 40.85 % | 2.17 |
| 0.01 | 0.01 | **3.57** | 4.82 | **8.63** | 18.87 | 42.63 % | 2.22 |

Table 4: **Analysis on different training schemes.** "Chairs" represents the FlyingChairs dataset and "KITTI" denotes the KITTI raw dataset. The baseline semi-supervised settings cannot improve the flow accuracy as the brightness constancy assumption does not hold on occluded regions. In contrast, our approach effectively utilizes the unlabeled data to improve the performance.

| Method | Training Datasets | Sintel-Clean EPE | Sintel-Final EPE | KITTI 2012 EPE | KITTI 2015 EPE | KITTI 2015 Fl | FlyingChairs EPE |
|---|---|---|---|---|---|---|---|
| Supervised | Chairs | 3.51 | 4.70 | 7.69 | 17.19 | 40.82% | 2.15 |
| Unsupervised | KITTI | 8.01 | 8.97 | 16.54 | 25.53 | 54.40% | 6.66 |
| Baseline semi-supervised | Chairs + KITTI | 3.69 | 4.86 | 10.38 | 18.07 | 39.33% | 2.11 |
| Proposed semi-supervised | Chairs + KITTI | **3.30** | **4.68** | **7.16** | **16.02** | **38.77**% | **1.95** |

of labeled data with ground truth flow from the FlyingChairs dataset and treat the remaining part as unlabeled data to train the proposed semi-supervised method. We also train the purely supervised method with the same amount of labeled data for comparisons. Table 5 shows that the proposed semi-supervised method consistently outperforms the purely supervised method on the Sintel, KITTI2012 and KITTI2015 datasets. The performance gap becomes larger when using less labeled data, which demonstrates the capability of the proposed method on utilizing the unlabeled data.

## 4.3 Comparisons with the state-of-the-arts

In Table 6, we compare the proposed algorithm with four variational methods: EpicFlow [32], DeepFlow [39], LDOF [4] and FlowField [2], and four CNN-based algorithms: FlowNetS [8], FlowNetC [8], SPyNet [30] and FlowNet 2.0 [16]. We further fine-tune our model on the Sintel training set (denoted by "+ft") and compare with the fine-tuned results of FlowNetS, FlowNetC, SPyNet, and FlowNet2. We note that the SPyNet+ft is also fine-tuned on the Driving dataset [26] for evaluating on the KITTI2012 and KITTI2015 datasets, while other methods are fine-tuned on the Sintel training data. The FlowNet 2.0 has significantly more network parameters and uses more training datasets (e.g., FlyingThings3D [26]) to achieve the state-of-the-art performance. We show that our model achieves competitive performance with the FlowNet and SPyNet when using the same amount of ground truth flow (i.e., FlyingChairs and Sintel datasets). We present more qualitative comparisons with the state-of-the-art methods in the supplementary material.

## 4.4 Limitations

As the images in the KITTI raw dataset are captured in driving scenes and have a strong prior of forward camera motion, the gain of our semi-supervised learning over the supervised setting is mainly on the KITTI 2012 and 2015 datasets. In contrast, the Sintel dataset typically has moving objects with various types of motion. Exploring different types of video datasets, e.g., UCF101 [35] or DAVIS [29], as the source of unlabeled data in our semi-supervised learning framework is a promising future direction to improve the accuracy on general scenes.

Table 5: **Training on partial labeled data.** We use $75\%$, $50\%$ and $25\%$ of data with ground truth flow from the FlyingChair dataset as labeled data and treat the remaining part as unlabeled data. The proposed semi-supervised method consistently outperforms the purely supervised method.

| Method | Amount of labeled data | Sintel-Clean EPE | Sintel-Final EPE | KITTI 2012 EPE | KITTI 2015 EPE | KITTI 2015 Fl-all | FlyingChairs EPE |
|---|---|---|---|---|---|---|---|
| Supervised | 75% | 4.35 | 5.40 | 8.22 | 17.43 | 41.62% | 1.96 |
| Proposed semi-supervised | 75% | 3.58 | 4.81 | 7.30 | 16.46 | 41.00% | 2.20 |
| Supervised | 50% | 4.48 | 5.46 | 9.34 | 18.71 | 42.14% | 2.04 |
| Proposed semi-supervised | 50% | 3.67 | 4.92 | 7.39 | 16.64 | 40.48% | 2.28 |
| Supervised | 25% | 4.91 | 5.78 | 10.60 | 19.90 | 43.79% | 2.09 |
| Proposed semi-supervised | 25% | 3.95 | 5.00 | 7.40 | 16.61 | 40.68% | 2.33 |

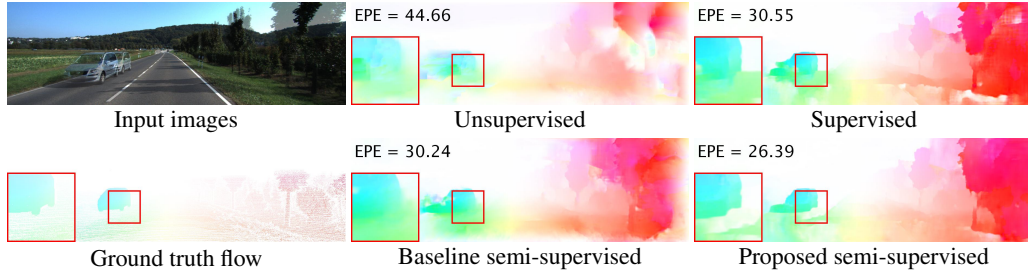

EPE = 44.66    EPE = 30.55

Input images        Unsupervised         Supervised

EPE = 30.24    EPE = 26.39

Ground truth flow    Baseline semi-supervised    Proposed semi-supervised

Figure 5: **Comparisons of training schemes.** The proposed method learns the flow warp error using the adversarial training and improve the flow accuracy on motion boundary.

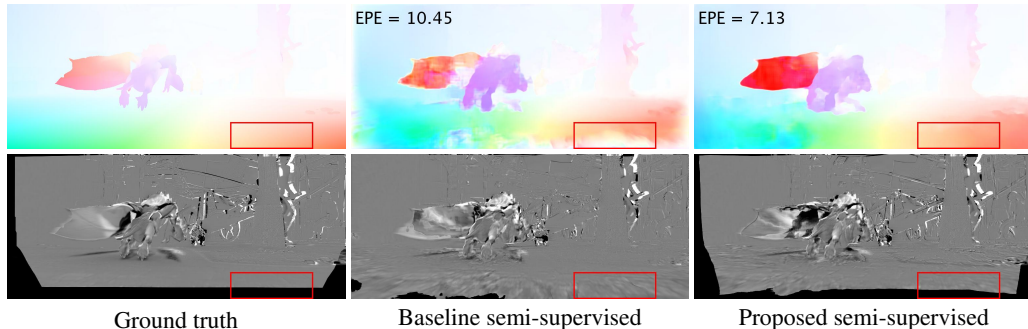

EPE = 10.45    EPE = 7.13

Ground truth        Baseline semi-supervised    Proposed semi-supervised

Figure 6: **Comparisons of flow warp error.** The baseline semi-supervised approach penalizes the flow warp error on occluded regions and thus produce inaccurate flow.

Table 6: **Comparisons with state-of-the-arts.** We report the average EPE on six benchmark datasets and the Fl score on the KITTI 2015 dataset.

| Method | | Middlebury Train EPE | Sintel-Clean Train EPE | Test EPE | Sintel-Final Train EPE | Test EPE | KITTI 2012 Train EPE | Test EPE | KITTI 2015 Train EPE | Train Fl-all | Test Fl-all | Chairs Test EPE |
|---|---|---|---|---|---|---|---|---|---|---|---|---|
| EpicFlow | [32] | 0.31 | 2.27 | 4.12 | 3.57 | 6.29 | 3.47 | 3.8 | 9.27 | 27.18% | 27.10% | 2.94 |
| DeepFlow | [39] | 0.25 | 2.66 | 5.38 | 4.40 | 7.21 | 4.58 | 5.8 | 10.63 | 26.52% | 29.18% | 3.53 |
| LDOF | [4] | 0.44 | 4.64 | 7.56 | 5.96 | 9.12 | 10.94 | 12.4 | 18.19 | 38.11% | - | 3.47 |
| FlowField | [2] | 0.27 | 1.86 | 3.75 | 3.06 | 5.81 | 3.33 | 3.5 | 8.33 | 24.43% | - | - |
| FlowNetS | [8] | 1.09 | 4.50 | 7.42 | 5.45 | 8.43 | 8.26 | - | 15.44 | 52.86% | - | 2.71 |
| FlowNetC | [8] | 1.15 | 4.31 | 7.28 | 5.87 | 8.81 | 9.35 | - | 12.52 | 47.93% | - | 2.19 |
| SpyNet | [30] | 0.33 | 4.12 | 6.69 | 5.57 | 8.43 | 9.12 | - | 20.56 | 44.78% | - | 2.63 |
| FlowNet2 | [16] | 0.35 | 2.02 | 3.96 | 3.14 | 6.02 | 4.09 | - | 10.06 | 30.37% | - | 1.68 |
| FlowNetS + ft | [8] | 0.98 | (3.66) | 6.96 | (4.44) | 7.76 | 7.52 | 9.10 | - | - | - | 3.04 |
| FlowNetC + ft | [8] | 0.93 | (3.78) | 6.85 | (5.28) | 8.51 | 8.79 | - | - | - | - | 2.27 |
| SpyNet + ft | [30] | 0.33 | (3.17) | 6.64 | (4.32) | 8.36 | 4.13 | 4.7 | - | - | - | 3.07 |
| FlowNet2 + ft | [16] | 0.35 | (1.45) | 4.16 | (2.01) | 5.74 | 3.61 | - | 9.84 | 28.20% | - | - |
| Ours | | 0.37 | 3.30 | 6.28 | 4.68 | 7.61 | 7.16 | 7.5 | 16.02 | 38.77% | 39.71% | 1.95 |
| Ours + ft | | 0.32 | (2.41) | 6.27 | (3.16) | 7.31 | 5.23 | 6.8 | 14.69 | 30.30% | 31.01 % | 2.41 |

# 5 Conclusions

In this work, we propose a generative adversarial network for learning optical flow in a semi-supervised manner. We use a discriminative network and an adversarial loss to learn the structural patterns of the flow warp error without making assumptions on brightness constancy and spatial smoothness. The adversarial loss serves as guidance for estimating optical flow from both labeled and unlabeled datasets. Extensive evaluations on benchmark datasets validate the effect of the adversarial loss and demonstrate that the proposed method performs favorably against the purely supervised and the straightforward semi-supervised learning approaches for learning optical flow.

## Acknowledgement

This work is supported in part by the NSF CAREER Grant #1149783, gifts from Adobe and NVIDIA.

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
