[Reviews · NeurIPS 2017]

Reviewer 1



Summary: The paper presents a semi-supervised approach to learning optical flow using a generative adversarial network (GAN) on flow warp errors. Rather than using a handcrafted loss (e.g., deviation of brightness constancy + deviation from smoothness) the paper explores the use of a GAN applied to flow warp errors. Strengths: + novel semi-supervised approach to learning; some concerns on the novelty in the light of [21] + generally written well Weaknesses: - some key evaluations missing Comments: Supervised (e.g., [8]) and unsupervised (e.g., [39]) approaches to optical flow prediction have previously been investigated, the type of semi-supervised supervision proposed here appears novel. The main contribution is in the introduction of an adversarial loss for training rather than the particulars of the flow prediction architecture. As discussed in Sec. 2, [21] also proposes an adversarial scheme. A further comparative discussion is suggested. Is the main difference in the application domain? This needs to be clarified. Also the paper claims that "it is not making explicit assumptions on the brightness constancy". While the proposed approach does not use an Lp or robust loss, as considered in previous work, the concept of taking flow warp error images as input does in fact rely on a form of brightness constancy assumption. The deviation from prior work is not the assumption of brightness constancy but the actual loss. In reference to [39], it is unclear what is meant by the approach "requires significantly more sophisticated data augmentation techniques" since the augmentation follows [8] and is similar to that used in the current paper. The main weakness of the paper is in the evaluation. One of the main stated contributions (line 64) is that the proposed approach outperforms purely supervised approaches. Judging from Table 3, this may be a gross misstatement. For example, on Sintel-Clean FlowNet2 achieves 3.96 aEPE vs 6.28 by the proposed approach. Why weren't results reported on the KITTI test sets using the test server? Are the reported KITTI results including occluded regions (in other words all points) or limited to non-occluded regions? The limitations of the presented training procedure (Sec. 4.4) should have been presented much earlier. It would have been interesting to train the network with data all from the same domain, e.g., driving scenes. For instance, rather than train on FlyingChairs and KITTI raw, it would have been interesting to train on KITTI raw (no ground truth) and Virtual KITTI (with ground truth) and evaluate on KITTI TEST. Reported fine-tuned results should be presented for all approaches. Minor point: - line 22 "Due to the lack of the large-scale ground truth flow" --> "Due to the lack of large-scale ground truth flow" - line 32: "With the limited quantity and unrealistic of ground truth" Overall rating: The training approach itself appears to be novel (a more detailed comparative discussion with respect to [21] is suggested) and will be of interest to those working on flow and depth prediction. Based on the issues cited with the evaluation I cannot recommend acceptance at this point. Rebuttal: The reviewer acknowledges that the authors have addressed some of my concerns/suggestions regarding their evaluation. The performance is still below that of the state-of-the-art. Finetuning may help but has not been evaluated. Is the performance deficit, as suggested by the authors, due to the capacity of their chosen network or is it due to their proposed GAN approach? Would the actual training of a higher capacity network with the proposed adversarial loss be problematic? These are topics to be explored.

Reviewer 2



The paper proposes a semi-supervised learning scheme with GANs for optical flow. The proposed approach is reasonable, and some tiny improvements (Table 2 and 3) are achieved over fully supervised or baseline semi-supervised (similar) architectures. However, this is the case when the proposed approach uses all the train data of the supervised architectures and extra train data. I am missing a simple yet very relevant experiment where the Flying Chairs train data is split for supervised training and remaining "unlabeled" data. For example, the percent of supervised training can go from 10% to 100% and the results for both supervised and proposed semisupervised (that uses also the remaining part of the data as unlabeled) should be reported for all these partitions. In this way we can see how much flexibility brings the proposed approach wrt the fully supervised approach. Another experiments could mix the train data with Sintel or other sources as mentioned in section 4.4. Are there cases when using unlabeled data worsen the overall performance? such as it happens for KITTI 2012 when using the "baseline semi-supervised" with "Chairs+KITTI" in Table 2. Why?

Reviewer 3



This paper discusses the training of an optical flow estimation model, and proposes two things: 1) train the model in a semi-supervised way by using a supervised loss on labeled data, and an unsupervised loss on unlabeled data. 2) use the labeled data to also learn the unsupervised loss using a discriminator (in a GAN setting). I think this paper is interesting and well written. The different components which can be thought of as 3 different losses (supervised, hand-crafted-unsupervised and learned-unsupervised) are explained clearly in the text and the experimental setup explicitly demonstrates the benefits of each. Even though the experimental results are inferior to the state-of-the-art both in hand-crafted methods and learned methods, they shed light on how the ideas presented can be incorporated with better models and lead to better results. While the overall experimental setup seems to be fair, there is one issue that bothers me. I don’t understand how the baseline semi-supervised method can be worse than the supervised method. If \lambda is the weight of the unsupervised loss then the semi-supervised method can always converge to the supervised method by making \lambda close to zero. So was the \lambda not tuned to give the best result? Is it a generalization error? Perhaps seeing the training errors might help here. Minor issues: In the flow-warp definitions in line 148, I think it should be y = I_1 - W() . Typo in line 184/5 “effect of using different the receptive”. The numbers (mixed with equation numbers) in line 223-227 are confusing.